# A functional genetic toolbox for human tissue-derived organoids

Dawei Sun[1], Lewis Evans[2], Francesca Perrone[3], Vanesa Sokleva[3], Kyungtae Lim[1], Saba Rezakhani[4], Matthias Lutolf[4], Matthias Zilbauer[3], Emma L Rawlins[3]*

[1]Wellcome Trust/CRUK Gurdon Institute, University of Cambridge, Cambridge, United Kingdom; [2]Developmental Biology and Cancer Development, University College London, London, United Kingdom; [3]University of Cambridge, Cambridge, United Kingdom; [4]École Polytechnique Fédérale de Lausanne (EPFL), Lausanne, Switzerland

**Abstract** Human organoid systems recapitulate key features of organs offering platforms for modelling developmental biology and disease. Tissue-derived organoids have been widely used to study the impact of extrinsic niche factors on stem cells. However, they are rarely used to study endogenous gene function due to the lack of efficient gene manipulation tools. Previously, we established a human foetal lung organoid system (Nikolić et al., 2017). Here, using this organoid system as an example, we have systematically developed and optimised a complete genetic toolbox for use in tissue-derived organoids. This includes 'Organoid Easytag', our efficient workflow for targeting all types of gene loci through CRISPR-mediated homologous recombination followed by flow cytometry for enriching correctly targeted cells. Our toolbox also incorporates conditional gene knockdown or overexpression using tightly inducible CRISPR interference and CRISPR activation which is the first efficient application of these techniques to tissue-derived organoids. These tools will facilitate gene perturbation studies in tissue-derived organoids facilitating human disease modelling and providing a functional counterpart to many ongoing descriptive studies, such as the Human Cell Atlas Project.

*For correspondence:
elr21@cam.ac.uk

**Competing interest:** The authors declare that no competing interests exist.

## Main

CRISPR and its related techniques (CRISPR interference [CRISPRi] and CRISPR activation [CRISPRa]) have transformed the study of gene function in model systems. They have been rapidly adopted in cancer and pluripotent stem cell (PSC) lines (*Bowden et al., 2020*; *Gilbert et al., 2013*; *Tian et al., 2019*), but not in tissue-derived human organoids. We have optimised these genetic tools for use in organoids using a tissue-derived human foetal lung organoid system (*Nikolić et al., 2017*).

First, we aimed to establish a robust workflow for gene targeting in organoids to facilitate reporter and direct knockout (KO) generation. A recent application of non-homologous end joining (NHEJ) to improve gene targeting in organoids has been successful (*Artegiani et al., 2020*). However, we adopted a homology-directed repair (HDR) strategy as a complementary approach. We reasoned that the recombination-based method allows our strategy to deliver precise genetic manipulation with flexible targeting sites and minimal additional genetic changes. We chose fluorescence as a selection marker, allowing targeted cells to be easily isolated using flow cytometry and chimeric colonies to be identified and removed using a fluorescent microscope.

To achieve efficient gene targeting, we first sought to maximise (1) the efficiency of DNA delivery into organoid cells and (2) the efficiency of site-specific DNA cleavage by the Cas9-gRNA complex. Nucleofection achieved up to 70% transfection efficiency and showed consistency across different organoid lines (*Figure 1—figure supplement 1a and b*). To optimise site-specific DNA cleavage, we

used nucleofection to introduce the Cas9-gRNA complex into cells in different forms (*Figure 1—figure supplement 1c*). Consistent with previous reports, Cas9 ribonucleoproteins (RNPs) outperformed plasmid-based Cas9 approaches (*Kim et al., 2014*; *Lin et al., 2014*), both in the T7 endonuclease assay and an online CRISPR editing analysis tool (*Figure 1—figure supplement 1d*). Thus, we adopted nucleofection and single-stranded synthetic gRNA combined with an RNP (ssRNP) for downstream experiments. This strategy has the advantage that the RNP is rapidly degraded and should produce minimal off-target effects.

To establish our gene targeting workflow, we first focused on generating a β-actin (ACTB)-fusion protein, taking advantage of the abundance of ACTB protein in human foetal lung organoids and a previously published targeting strategy (*Roberts et al., 2017*). We designed a repair template to generate an N terminal monomeric (m)*EGFP-ACTB* fusion based on the most efficient gRNA tested (*Figure 1a*). We set the following rules for repair template design to facilitate efficient and consistent gene targeting: (1) protospacer adjacent motif (PAM) sequence mutated to prevent editing by ssRNP (*Paquet et al., 2016*); (2) 700–1000 nt length of each homologous arm (*Yao et al., 2018*); (3) minimal plasmid size to maximise delivery into organoid cells (<7.0 kb in size recommended empirically) and (4) monomeric forms of fluorescent protein to avoid undesirable fusion protein aggregates. As expected, 72 hr after nucleofection of the ssRNP and repair template, mEGFP$^+$ organoid cells could be enriched by flow cytometry (*Figure 1b and c*). These cells were collected and pooled together, but seeded sparsely, and successfully expanded into organoid colonies (*Figure 1d*). The mEGFP-ACTB fusion protein localised to cell–cell junctions as expected (*Roberts et al., 2017*). These small colonies could be further expanded into new organoid lines and 59% of lines (n = 17/29 lines, from N = 2 parental organoid lines, *Supplementary file 1*) were correctly targeted. Targeted organoids continued to express the multipotent lung progenitor marker, SOX9 (*Figure 1e*). We sought to further increase targeting efficiency using drugs previously reported to enhance HDR (*Maruyama et al., 2015*; *Song et al., 2016*; *Yu et al., 2015*). However, using flow cytometry as a simple assay, none of the drugs tested increased the rate of gene targeting (*Figure 1—figure supplement 2*).

The *AAVS1* locus (Adeno-Associated Virus Integration Site 1, located in the first intron of the *PPP1R12C* gene) has been considered to be a 'safe harbour locus' for expressing exogenous genes in a controllable manner in human cells without silencing (*Smith et al., 2008*). As a further proof of concept, we successfully targeted *AAVS1* to express a membrane tagged TagRFP-T (mTagRFP-T) for cell shape visualisation (*Figure 1f and g*; n = 4/6 correctly targeted lines from N = 1 parental organoid line). Therefore, we have combined Cas9 RNP with single-stranded synthetic gRNA, a simple circular plasmid repair template and a strategy to enrich correctly targeted cells via flow cytometry for gene targeting in human foetal lung organoids. We named this workflow Organoid Easytag.

To expand our Organoid Easytag pipeline to reporter generation, we targeted *SOX9*. SOX9, a transcription factor, is a tip progenitor cell marker for developing lungs (*Nikolić et al., 2017*), and *SOX9* reporters are useful for monitoring organoid progenitor state. To overcome its low expression level, we used a Histone H2B-EGFP fusion (H2B-EGFP hereafter) to concentrate the EGFP signal in the nucleus (*Figure 2a*). A *T2A* sequence, a self-cleavage peptide, was also inserted between *SOX9* and *H2B-EGFP* to ensure that SOX9 protein was minimally influenced. This strategy allowed us to enrich correctly targeted cells by flow cytometry. Targeted colonies could be expanded and maintained normal SOX2, SOX9 and NKX2-1 expression (*Figure 2b*, *Figure 2—figure supplement 1a*; n = 9/23 correctly targeted lines from N = 3 parental organoid lines). Importantly, we noted that although we were only able to generate SOX9 reporter lines as heterozygotes (*Figure 2—figure supplement 1b and c*), the gRNA sites in the wildtype alleles were intact (6/6 lines tested, N = 3 parental organoid lines) (*Figure 2—figure supplement 1d*). This offers the opportunity to retarget the second allele if desired. To evaluate potential off-target effects, we selected the top nine most probable *SOX9* gRNA off-target sites (*Figure 2—figure supplement 1e*) and tested if these regions contained any undesired CRISPR-induced scars. No indels were detected within ~200 bps flanking these potential off-target sites (*Figure 2—figure supplement 1e and f*), suggesting low off-target effects of our workflow (3/3 *SOX9-T2A-H2B-EGFP* lines tested). Additionally, we validated that the *SOX9*-targeted organoids retained the potential to differentiate (*Figure 2—figure supplement 1g*), further demonstrating the normality of the targeted organoid lines.

We sought to apply Organoid Easytag to transcriptionally silent loci to generate differentiation reporters. We adopted the strategy of inserting an exogenous promoter flanked by two Rox sites

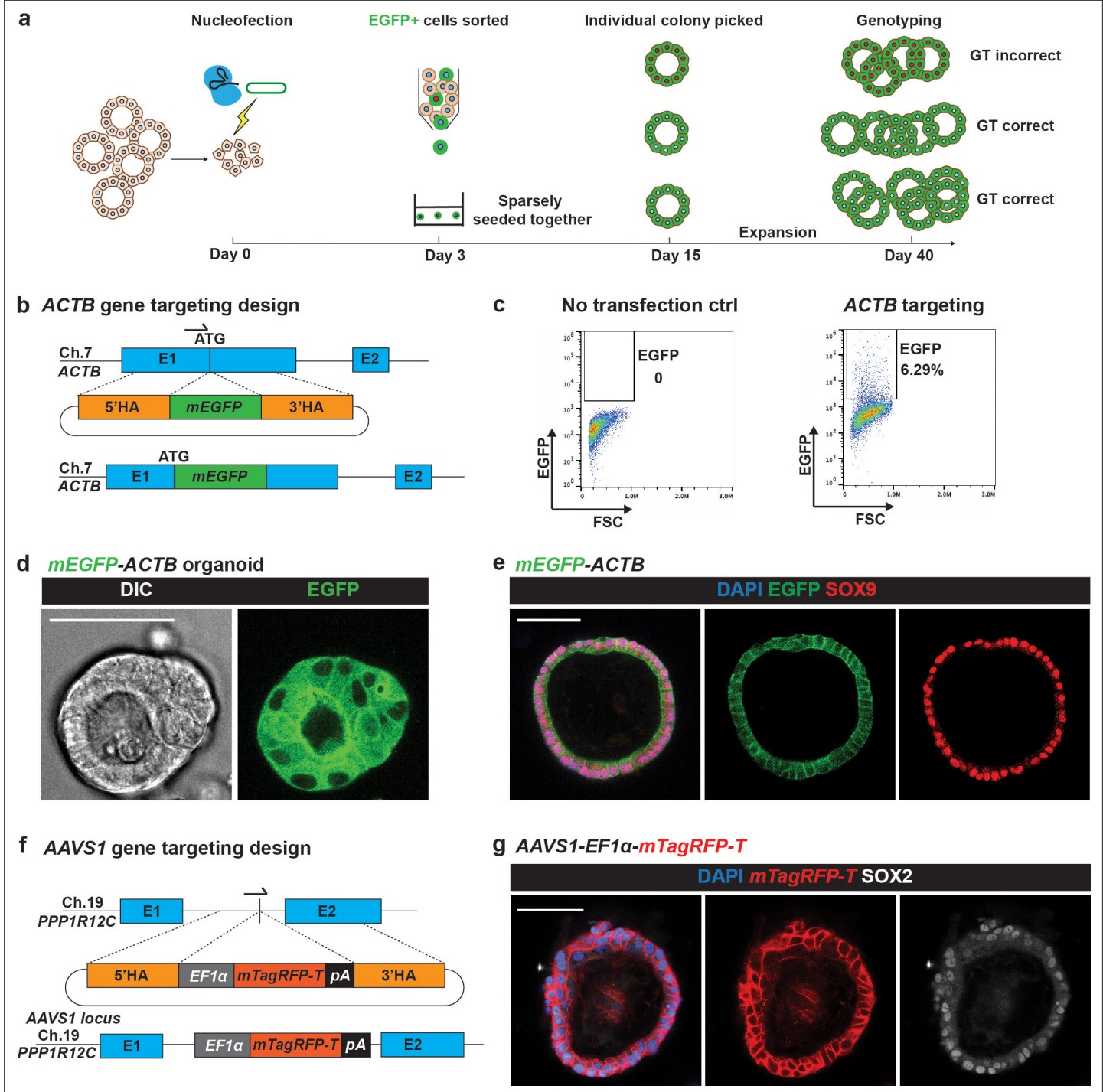

**Figure 1.** The Organoid Easytag workflow for gene targeting in human organoids. (**a**) Schematic of the Organoid Easytag workflow. ssRNP and a circular plasmid repair template are nucleofected into dissociated cells at day 0. By day 3, cells have proliferated to become tiny colonies and are removed from the Matrigel and dissociated for selection by flow cytometry. EGFP$^+$ cells are re-plated sparsely (~1000–1500 cells/well of a 24-well plate) and grown until day 15 when organoids reach a sufficient size to be manually picked under a fluorescent microscope. Typically, 10–40 organoid colonies formed per ~1000 cells seeded. Organoids are picked into individual wells and passaged until sufficient cells are obtained for both genotyping and freezing down the line. Cells with red nuclei represent incorrectly targeted cells. Cells with white nuclei denote correctly targeted cells. (**b**) Schematic of repair template design for N terminal fusion *mEGFP-ACTB* gene targeting and the final product. Arrow shows the position of gRNA. E1, exon 1; E2, exon 2; 5'HA, 5' homology arm; 3'HA, 3' homology arm. (**c**) Representative flow cytometry results showing the percentage of EGFP cells 72 hr after nucleofection is performed. (**d**) Representative image showing *mEGFP-ACTB* organoid. DIC channel on the left and EGFP channel on the right. (**e**) Immunofluorescence of *mEGFP-ACTB* organoids. Blue: DAPI (nuclei); green: EGFP (ACTB fusion protein); red: SOX9 (lung progenitor marker).

*Figure 1 continued on next page*

*Figure 1 continued*

(**f**) Schematic of the *AAVS1* targeting repair template design and the final product. E1, exon 1, E2, exon 2. Arrow indicates the position of the gRNA. (**g**) Immunofluorescence of *AAVS1-EF1a-mTagRFP-T* organoids. Blue: DAPI (nuclei); red: mTagRFP-T (membrane localised reporter); white: SOX2 (lung progenitor marker). Scale bars in all panels denote 50 μm.

The online version of this article includes the following figure supplement(s) for figure 1:

**Figure supplement 1.** Optimisation of DNA delivery and CRISPR cutting.

**Figure supplement 2.** Small molecules did not significantly improve organoid gene targeting efficiency.

which could be subsequently excised by Dre recombinase (*Anastassiadis et al., 2009*; *Roberts et al., 2019*). The exogenous *EF1a* promoter drives fluorescent reporter expression for flow cytometry selection, allowing us to first positively enrich correctly targeted cells (Venus⁺), and subsequently enrich (Venus⁻) cells following Dre-mediated *EF1a* removal (*Figure 2—figure supplement 2a*). This design also helps to minimise the repair template size. We targeted the *SFTPC* (surfactant protein C, *Figure 2—figure supplements 2–3*) and *TP63* genes (*Figure 2—figure supplement 4*) because they are not expressed in human foetal lung organoid cells and are well-established markers for alveolar type II and basal cell lineages, respectively (*Barkauskas et al., 2013*; *Rock et al., 2009*). To test *SFTPC* reporter function, we overexpressed NKX2-1 from a lentiviral vector (*Figure 2—figure supplement 2d*; *Lim et al., 2021*). Following NKX2-1 induction, Venus was co-expressed with proSFTPC protein, confirming that the reporter is functional (*Figure 2—figure supplement 2e*). We noted that the Venus signal was cytoplasmic, likely due to inefficient self-cleavage of the T2A sequence (*Figure 2—figure supplement 2f*). *TP63* reporter organoid lines were produced using the same strategy (*Figure 2—figure supplement 4*). These results indicate that the Organoid Easytag workflow can target silent gene loci.

The generation of straightforward gene KOs by induction of indels using the CRISPR-Cas9 system can suffer from translation retention and exon skipping (*Smits et al., 2019*; *Tuladhar et al., 2019*). Moreover, in the absence of a strong, immediate phenotype the KO cells cannot readily be identified. We sought to solve these problems by generating a gene KO in a controlled manner using Organoid Easytag. We focused on *SOX2* as its function remains to be elucidated in human foetal lung progenitors. We replaced the *SOX2* coding sequence (CDS) with *T2A-H2B-EGFP* to generate *SOX2* KO organoids. Using two gRNAs targeting the N and C terminal of the *SOX2* CDS, respectively, we sequentially replaced both copies of the *SOX2* CDS (*Figure 2c*, *Figure 2—figure supplement 5*). The EGFP signal was increased after targeting the second copy of the *SOX2* CDS, making selection of serially targeted alleles by flow cytometry straightforward (*Figure 2—figure supplement 5c*). *SOX2* KO colonies proliferate and grow normally for at least four serial passages, suggesting that SOX2 is not crucial for human foetal lung tip progenitor cell self-renewal (*Figure 2d*).

Additionally, we tested the Organoid Easytag workflow in the human foetal intestinal organoid system (*Elmentaite et al., 2020*) and successfully derived *SOX9* reporter lines (*Figure 2e and f*; 2/4 lines tested correctly targeted).

We have demonstrated that our Organoid Easytag pipeline can target various loci, including highly abundant genes, transcription factors, the human safe harbour locus and transcriptionally silent genes. The Organoid Easytag workflow can also be adapted to generate KOs and applied to different organoid systems (efficiency details summarised in *Supplementary file 1*). Organoid Easytag will be a complementary technique to the recently published CRISPR-HOT approach (*Artegiani et al., 2020*) and is particularly suited for experiments where precise gene editing is required (*Figure 2—figure supplement 5f*).

KOs are not suitable for studying the function of genes which are required for stem cell self-renewal. Moreover, temporal, and reversible, control of gene expression cannot be easily achieved using KOs. Inducible CRISPRi could potentially solve these problems. We first embedded an N-terminal *KRAB-dCas9* fusion (*Mandegar et al., 2016*) in a doxycycline (Dox)-inducible TetON system aiming to gain temporal control of CRISPRi function. However, the TetON system alone did not provide sufficiently tight control of CRISPRi, possibly due to copy number or integration position of the transgene (*Figure 3—figure supplement 1*). To solve this problem, we fused KRAB-dCas9 with a destabilising domain derived from *Escherichia coli* dihydrofolate reductase (DHFR) (*Figure 3a*). DHFR is stabilised by a small molecule, trimethoprim (TMP) (*Figure 3b*; *Iwamoto et al., 2010*). We evaluated the new

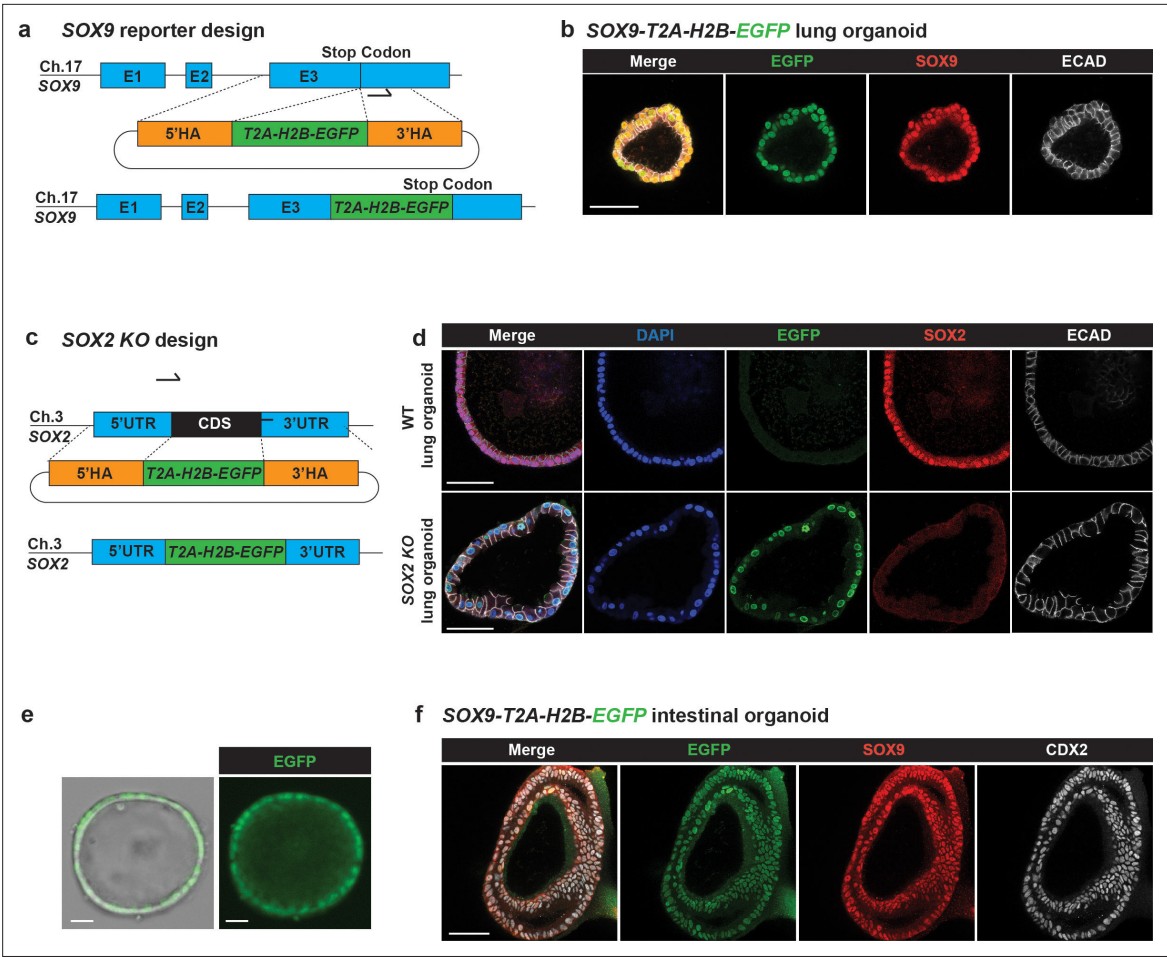

**Figure 2.** The Organoid Easytag workflow can be adapted for multiple applications. (**a**) Schematic of the *SOX9* locus repair template design and final product. E1, exon 1; E2, exon 2; E3, exon 3; 5' HA, 5' homology arm; 3' HA, 3' homology arm. Arrow shows the position of the gRNA. (**b**) Immunofluorescence of *SOX9-T2A-H2B-EGFP* human foetal lung organoids. Green: EGFP (SOX9 reporter); red: SOX9 (lung progenitor marker); white: ECAD (E-cadherin, basal-lateral junctions). (**c**) Schematic showing repair template design and final product for the generation of *SOX2* knockout organoids using Organoid Easytag. Two gRNA sites were used at the N and C terminal of the *SOX2* CDS, respectively. *SOX2* CDS was replaced by *T2A-H2B-EGFP*. (**d**) Representative immunofluorescence showing that SOX2 protein is completely knocked out. Blue: DAPI (nuclei); green: EGFP (*SOX2* transcriptional reporter); red: SOX2 protein (lung progenitor marker); white: ECAD (E-cadherin, basal-lateral junctions). (**e**) Widefield microscopic images showing *SOX9* reporter human foetal intestinal organoid. (**f**) Immunofluorescence of *SOX9-T2A-H2B-EGFP* human foetal intestinal lung organoids. Green: EGFP (SOX9 reporter); red: SOX9; white: CDX2 (intestinal lineage marker). Scale bars denote 50 µm.

The online version of this article includes the following figure supplement(s) for figure 2:

**Figure supplement 1.** Characterisation of *SOX9* targeted colonies.

**Figure supplement 2.** Organoid Easytag to target silent gene, *SFTPC* locus.

**Figure supplement 3.** Characterisation of *SFTPC* reporter organoids.

**Figure supplement 4.** Organoid Easytag to target the *TP63* locus.

**Figure supplement 5.** Generation of *SOX2* knockout using Organoid Easytag workflow.

inducible CRISPRi system by depleting a ubiquitous cell surface marker CD71 (transferrin receptor C [TFRC]). Using previously validated gRNAs (*Horlbeck et al., 2016*), CD71 protein was depleted in the majority of the cells after 5 days of Dox and TMP treatment (91.4% ± 2.1%, mean ± SEM, N = 3; *Figure 3c and d*; *Figure 3—figure supplement 2a and b*). Whereas no knockdown was observed in the no Dox/TMP treatment group, or in organoids transduced with non-targeting control gRNA. The knockdown effect was achieved after 5 days, could be reversed after 1–2 weeks of Dox and TMP removal and then induced again (*Figure 3—figure supplement 2a–c*), consistent with a recent report (*Nuñez et al., 2021*). We further evaluated the inducible CRISPRi system using *SOX2*. SOX2

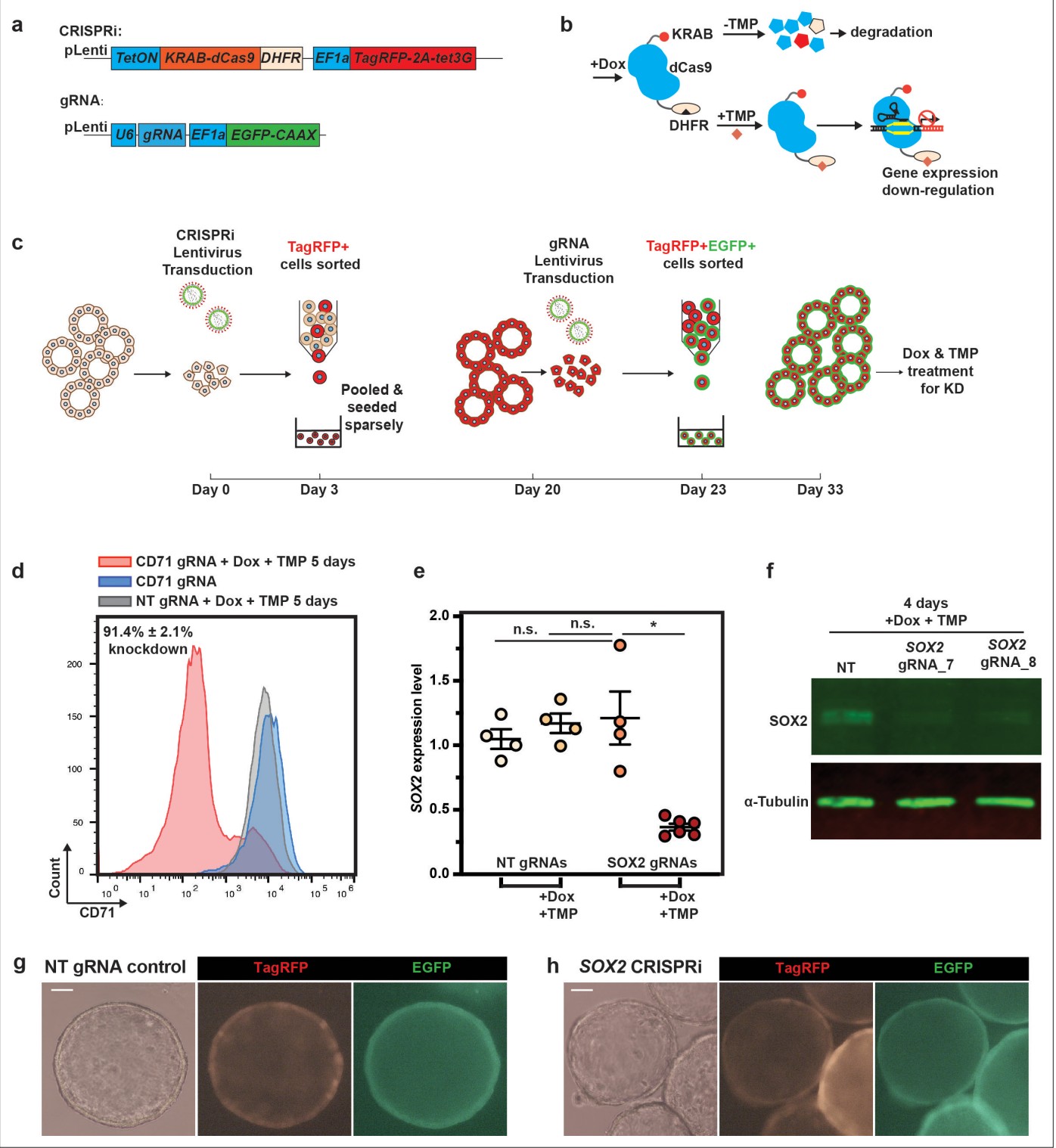

**Figure 3.** Precise temporal control of CRISPRi in human foetal lung organoids can be achieved via combining the doxycycline (Dox)-inducible system with a protein destabilising domain (DD). (**a**) Schematic of lentiviral vector designs for sequential introduction of inducible CRISPRi and gRNAs into human foetal lung organoids. N-terminal KRAB-dCas9 was fused with dihydrofolate reductase (DHFR)-derived DD for stringent control of CRISPRi function. EGFP was fused with a C-terminal CAAX domain to achieve membrane labelling of the transduced cells. (**b**) Schematic showing how the temporal control of CRISPRi protein is achieved by combining the Dox-inducible system and DD. The Dox-inducible system is somewhat leaky, but in the absence of trimethoprim (TMP), any KRAB-dCas9-DHFR fusion protein produced is degraded. Dox treatment results in high levels of KRAB-dCas9-DHFR fusion protein expression. When TMP is not present, the fusion protein is destabilised, whereas when TMP is supplemented, the fusion protein

*Figure 3 continued on next page*

*Figure 3 continued*

is stabilised and can exert its gene expression repression function. (**c**) Workflow for generating gene knockdown (KD) using CRISPRi system. A parental organoid line with inducible CRISPRi was produced via lentiviral transduction followed by sorting for TagRFP-positive cells. Single cells were re-plated (~3000–5000 cells/well of a 24-well plate) and expanded for around 17 days. gRNA lentivirus was then introduced by a second lentiviral transduction event, followed by sorting for TagRFP/EGFP dual-positive cells after 3 days. Cells were re-plated (~2000–3000 cells/well of a 24-well plate) and expanded for another 10 days before treatment with Dox and TMP. 4–6 days after Dox and TMP treatment, gRNA performance was evaluated. (**d**) Representative flow cytometry results for validation of inducible CRISPRi performance using CD71 as a target. After TMP and Dox administration for 5 days, CD71 protein was successfully downregulated. Grey histogram indicates CD71 expression level for organoids with non-targeting (NT) control gRNA after Dox and TMP treatment for 5 days; blue histogram indicates the CD71 expression level for organoids with CD71 gRNA without Dox or TMP treatment, showing no KD effect; pink histogram indicates the CD71 expression level for CD71 gRNA with Dox and TMP treatment for 5 days. The figures on the graph indicate the percentage of CD71$^{low}$ cells, mean ± SEM. Two independent organoid lines with two different CD71 gRNAs were used. (**e**) *SOX2* can be effectively knocked down using inducible CRISPRi without leaky effects indicated by qRT-PCR. Organoids with Dox and TMP treatment were harvested 5 days after the treatment. Each dot represents an independent experiment. Two independent organoid lines with two different NT control gRNAs and three different SOX2 gRNAs were used. *SOX2* expression level was normalised to organoids with NT control gRNAs. Detailed breakdown of these data is shown in *Figure 4—figure supplement 1*. Error bars are plotted to show mean ± SEM. A two-sided Student's t-test with non-equal variance was used to compare the means of each group. * indicates p-value < 0.05. n.s. indicates non-significant. (**f**) Representative western blot results showing SOX2 was effectively knocked down using the inducible CRISPRi system (gRNA7 and gRNA8 are independent guides targeted to *SOX2*; N = 2 parental organoid lines used). (**g–, h**) Representative images showing organoid morphology after *SOX2* knockdown. Images were taken after 5 days of Dox and TMP treatment. *SOX2* knockdown (right panel) did not result in obvious phenotypic changes in organoids compared with NT gRNA control (left panel). Scale bars denote 100 μm.

The online version of this article includes the following figure supplement(s) for figure 3:

**Figure supplement 1.** The tetON system alone was not able to tightly control CRISPRi function.

**Figure supplement 2.** Kinetics and reversibility of the inducible CRISPRi system.

could be efficiently knocked down at both the transcriptional and protein level and no background CRISPRi function was observed (*Figure 3e and f*). The *SOX2* knockdown occurred by day 1 and was also reversible (*Figure 3—figure supplement 2d*), indicating that the CRISPRi function is efficient and tightly regulated. Consistent with our *SOX2* KO, we did not observe effects on organoid morphology or growth after *SOX2* knockdown (*Figure 3g and h*). This further confirmed our finding that SOX2 is not crucial for organoid self-renewal.

Finally, we have assembled the CRISPRa system to switch on endogenous genes in human foetal lung organoids. We tested a previously reported CRISPRa system, dxCas9(3.7)-VPR (*Hu et al., 2018*), for its ability to activate *SFTPC* transcription (*Figure 4*). Inducible CRISPRa can induce *SFTPC* gene expression efficiently at both transcriptional and protein level without influencing organoid growth (*Figure 4b–d*).

We have established a complete organoid genetic toolbox for gene targeting, reporter generation, controllable gene KOs, inducible gene knockdown and gene activation in the human foetal lung organoid system (efficiency details provided in *Supplementary file 1*). We envision that these tools can be easily adapted for organoid systems derived from other tissues that are suitable to be nucleofected, or transduced by lentivirus. This will empower tissue-derived organoids to model human congenital lung disease and benchmark gene function using the Human Cell Atlas as a reference.

## Materials and methods
### Human embryonic and foetal lung tissue

Human embryonic and foetal lungs were obtained from terminations of pregnancy from Cambridge University Hospitals NHS Foundation Trust under permission from the NHS Research Ethical Committee (96/085) and the Joint MRC/Wellcome Trust Human Developmental Biology Resource (London and Newcastle, http://www.hdbr.org/) under permission from the NHS Research Ethics Committee (18/LO/0822 and 18/NE/0290). Samples used had no known genetic abnormalities.

### Derivation and maintenance of human foetal lung organoid culture

Human foetal lung organoids were derived and maintained as previously reported (*Nikolić et al., 2017*). Briefly, human foetal lung tissues were dissociated using Dispase (8 U/ml, Thermo Fisher Scientific, 17105041) at room temperature (RT) for 2 min. Mesenchyme was dissected away using needles. Tips of the branching epithelium were micro-dissected, transferred into 50 μl of Matrigel (356231,

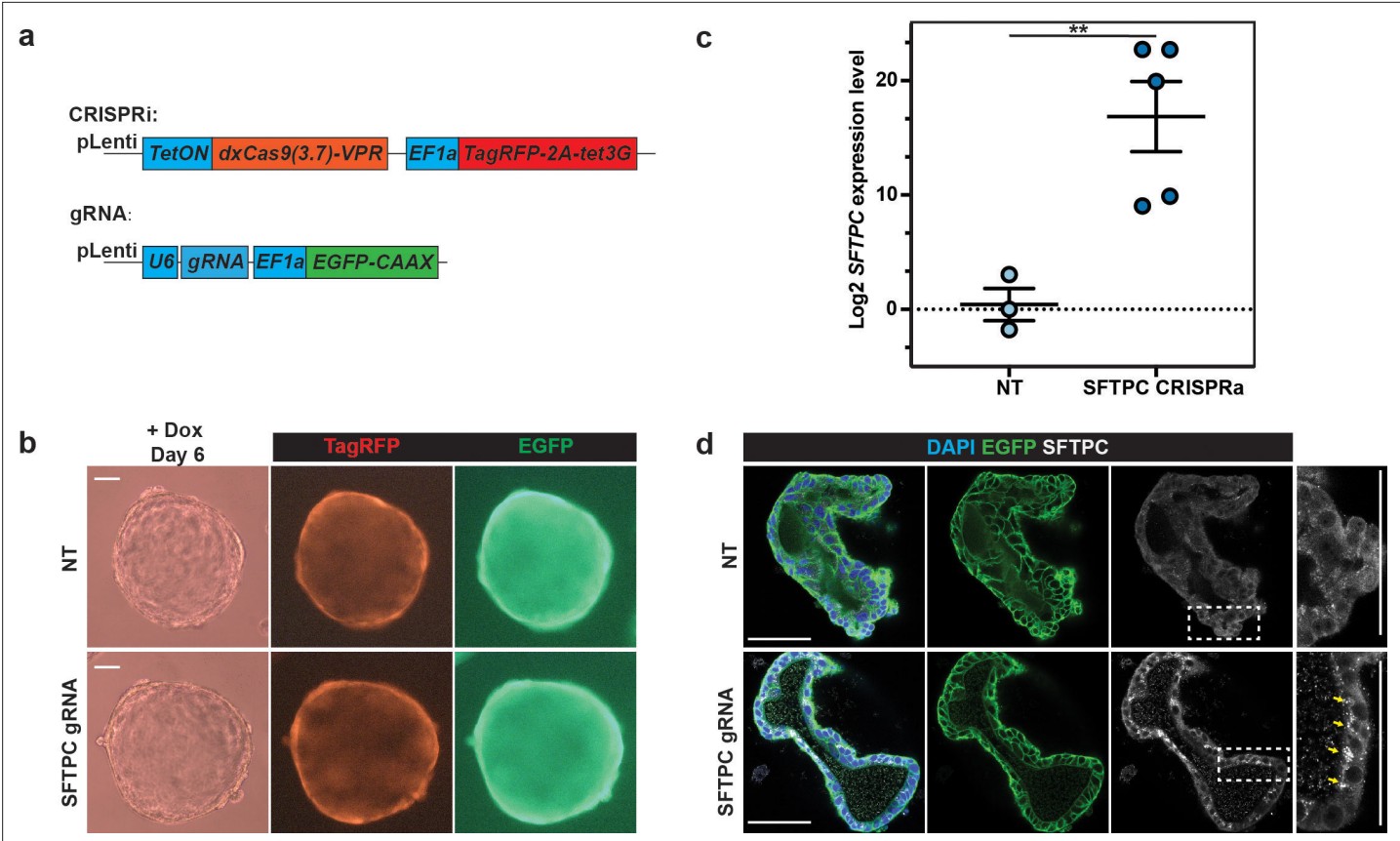

**Figure 4.** Inducible CRISPRa drove expression of a silent gene, *SFTPC*, in human foetal lung organoids. (**a**) Schematic of lentiviral design for introducing the CRISPRa system and gRNA into human foetal lung organoids. (**b**) Representative images to show organoid morphology after *SFTPC* activation after 6 days of doxycycline (Dox) treatment. *SFTPC* activation (lower panel) did not result in obvious phenotypic changes in organoids compared with non-targeting gRNA control (upper panel). (**c**) *SFTPC* can be effectively activated by CRISPRa system indicated by qRT-PCR. Each dot represents an independent experiment. Two independent organoid lines with two different non-targeting control (NT) gRNAs and three different *SFTPC* gRNAs were used. *SFTPC* expression level was normalised to organoids with non-targeting control (NT) gRNAs. A detailed breakdown of the results is given in *Figure 4—figure supplement 1*. Error bars are plotted to show mean ± SEM. A two-sided Student's t-test with non-equal variance was used to compare the means of each group. ** indicates p-value < 0.01. (**d**) Representative immunofluorescence showing proSFTPC protein accumulation in organoid cells after CRISPRa activation of *SFTPC* gene expression. Bright SFTPC protein aggregates accumulated in cells, consistent with the hydrophobic nature of SFTPC protein. Green: EGFP; white: proSFTPC protein. Scale bars denote 100 μm (**b**) and 50 μm (**d**).

The online version of this article includes the following figure supplement(s) for figure 4:

**Figure supplement 1.** *SOX2* CRISPRi and *SFTPC* CRISPRa results break down.

Corning) and seeded in one well of a 24-well low-attachment plate (M9312-100EA, Greiner). The plate was incubated at 37°C for 5 min to solidify the Matrigel. 600 μl of self-renewing medium containing N2 (1:100, Thermo Fisher Scientific, 17502-048), B27 (1:50, Thermo Fisher Scientific, 12587-010), N-acetylcysteine (1.25 mM, Merck, A9165), EGF (50 ng/ml, PeproTech, AF-100-15), FGF10 (100 ng/ml, PeproTech, 100-26), FGF7 (100 ng/ml, PeproTech, 100-19), Noggin (100 ng/ml, PeproTech, 120-10C ), R-spondin (5%, v/v, Stem Cell Institute, University of Cambridge), CHIR99021 (3 μM, Stem Cell Institute, University of Cambridge) and SB 431542 (10 μM, Bio-Techne, 1614) was added. The plate was incubated under standard tissue culture conditions (37°C , 5% $CO_2$). Once formed, organoids were maintained in self-renewing medium and passaged by mechanically breaking using P200 pipettes every 10–14 days.

## Whole-mount immunostaining for human foetal lung organoids

Organoids were fixed with 4% paraformaldehyde (PFA) directly in the culture plates on ice for 30 min. After two PBS washes, 0.5% (w/v) bovine serum albumin (BSA), 0.2% Triton-X in PBS (washing solution) was added and left on ice overnight to dissolve Matrigel. Organoids were then transferred into

multiple CellCarrier-96 Ultra Microplates (PerkinElmer, 6055300) for staining. Subsequently, blocking was performed in 5% donkey serum (Stratech, 017-000-121-JIR), 0.5% (w/v) BSA, 0.2% Triton-X in PBS (blocking solution) at 4°C overnight. For primary antibody staining, the following antibodies in blocking solution were used at 4°C overnight: SOX2 (1:500, Bio-Techne, AF2018, RRID:AB_355110), SOX9 (1:500, Merck, AB5535, RRID:AB_2239761), E-cadherin (1:1500, Thermo Fisher Scientific, 13-1900, RRID:AB_2533005), NKX2-1 (1:500, Abcam, ab76013, RRID:AB_1310784), TagRFP (1:1000, Evrogen, AB233, RRID:AB_2571743), GFP/Venus (1:500, Abcam, ab13970, RRID:AB_300798), CDX2 (1:200, Abcam, ab157524, RRID:AB_2721036) and proSFTPC (1:500, Merck, AB3786, RRID:AB_91588). After washing off the primary antibodies, the following secondary antibodies in washing buffer were used at 4°C overnight: donkey anti-chick Alexa 488 (1:2000, Jackson Immune, 703-545-155, RRID:AB_2340375), donkey anti-rabbit Alexa 594 (1:2000, Thermo Fisher Scientific, A-21207, RRID:AB_141637), donkey anti-goat Alexa 594 (1:2000, Thermo Fisher Scientific, A-11058, RRID:AB_2534105), donkey anti-goat Alexa 647 (1:2000, Thermo Fisher Scientific, A-21447, RRID:AB_2535864), donkey anti-rat Alexa 647 (1:2000, Jackson Immune, 712-605-153, RRID:AB_2340694) and donkey anti-mouse Alexa 647 (1:2000, Thermo Fisher Scientific, A31571, RRID:AB_162542). The following day, DAPI (Sigma, D9542) staining was performed in washing solution at 4°C for 30 min. After two washes with PBS, 97% (v/v) 2'-2'-thio-diethanol (TDE, Merck, 166782) in PBS was used for mounting. Confocal z stacks were acquired using Leica SP8 at an optical resolution of 1024 × 1024 using a ×40 lens. Single-plane images are shown. Images were processed using ImageJ (version 2.0.0).

## Plasmid nucleofection

For testing transduction efficiency, organoids were dissociated into single cells using pre-warmed TrypLE Express (12605028, Thermo Fisher Scientific) at 37°C for 10 min. The reaction was terminated by adding Advanced DMEM/F12 (12634028, Thermo Fisher Scientific) and cells passed through a 30 µm cell strainer. $2 \times 10^5$ organoid single cells were re-suspended with Lonza P3 nucleofection buffer and 1 µl of pmaxGFP (Lonza) and transferred to 20 µl nucleofection cuvette (V4XP-3024, Lonza). Nucleofection was performed using Lonza 4D Nucleofector with X unit using the program EA125. After nucleofection, self-renewing medium supplemented with 10 µM Y-27632 (ROCK inhibitor, ROCKi, 688000, Merck) was added to dilute the P3 buffer. Cell mixture was then seeded in Matrigel in two wells of a 24-well plate and cultured with self-renewing medium with ROCKi (10 µM) for 72 hr before FACS analysis.

## Lentiviral production

HEK293T cells were kindly provided by Dr. Rick Livesey and tested mycoplasma negative. We grew HEK293T cells in 10 cm dishes to a confluence of 80% before we transfected the lentiviral vector (10 µg) with packaging vectors including pMD2.G (3 µg, Addgene plasmid # 12259), psPAX2 (6 µg, Addgene plasmid # 12260) and pAdVAntage (3 µg, E1711, Promega) using Lipofectamine 2000 Transfection Reagent (11668019, Thermo Fisher Scientific) according to the manufacturer's protocol. After 16 hr, medium was refreshed. Supernatant containing lentivirus was harvested at 24 hr and 48 hr after medium refreshing and pooled together. Supernatant was centrifuged at 300 g to remove cell fragments and passed through 0.45 µm filter. Lentivirus containing >10 kb length insert (inducible CRISPRi and CRISPRa) was concentrated using AVANTI J-30I centrifuge (Beckman Coulter) with JS-24.38 swing rotor at 72,000 g for 2 hr at 4°C and pellets were dissolved in 200 µl PBS. Other lentivirus, including gRNA and NKX2-1-inducible overexpression, were concentrated using Lenti-X Concentrator (631232, Takara) and pellets were dissolved in 400 µl PBS.

## Lentivirus infection of organoids to test infection efficiency

Organoid single-cell suspension was prepared as for nucleofection. 5 µl lentivirus (CMV-myrAKT-IRES-GFP) suspension was applied to $2 \times 10^5$ organoid single cells suspended in 500 µl self-renewing medium with 10 µM ROCKi (without Matrigel) in one well of 24-well plate and incubated at 37°C overnight. The following day, cells were harvested and washed twice with PBS before pelleting and seeding in Matrigel in two wells of a 24-well plate. Cells were grown in self-renewing medium with ROCKi (10 µM) for 72 hr before flow cytometry. CMV-myrAKT-IRES-GFP was only used for checking lentiviral transduction efficiency (*Supplementary file 1*).

## Lipofectamine transfection of organoids

Organoid single cells were prepared the same way as for nucleofection. For comparing transduction efficiency, 1 µg of pmaxGFP (Lonza) was mixed with 1 µl of Lipofectamine Stem Transfection Reagent (STEM00001, Thermo Fisher Scientific) according to the manufacturer's protocol. 50 µl reaction mixture was applied to $2 \times 10^5$ organoid single cells suspended with 450 µl self-renewing medium with ROCKi (without Matrigel) in a single well of a 24-well plate. The plate was then centrifuged at 32°C at 600 g for 1 hr, followed by incubation at 37°C for 2–4 hr. Cells were then harvested, pelleted and seeded in Matrigel in two wells of a 24-well plate and grown in self-renewing medium supplemented with ROCKi (10 µM) for 72 hr before FACS analysis.

## Nucleofection for gene targeting in human foetal lung organoids

Cas9 protein was prepared and used as previously reported (*Bruntraeger et al., 1961*). If synthetic single-strand gRNAs were used, 2 µl spCas9 (4 µg/µl) and 2.88 µl of single-strand synthetic RNA (ssRNA, 100 µM, Synthego) were mixed and incubated at RT for a minimum of 10 min in order to form ssRNPs. If synthetic crispr/tracer (cr/tr) RNA heterodimers were used, 200 pmol synthetic cr RNA (IDT) and 200 pmol synthetic tr RNA (IDT) were mixed with 2.5 µl Nuclease Free Duplex Buffer (11-01-03-01, IDT) and denatured at 95°C for 2 min. 2 µl of cr/tr RNA heterodimer was cooled down to RT on the bench, mixed with 2 µl spCas9 (4 µg/µl) and incubated at RT for a minimum of 10 min to form cr/tr RNPs. At the same time, organoids were dissociated into single cells, according to the protocol previously described for nucleofection. $4 \times 10^5$ cells were suspended using Lonza Nucleofection P3 buffer, mixed with 10 µg of appropriate plasmid repair template. The cell suspension was further mixed with pre-formed Cas9 RNPs and equally distributed into two 20 µl cuvettes (V4XP-3032, Lonza). Nucleofection was performed using program EA104 for human foetal lung organoids (EA125 could also be used, but resulted in more cell death). After nucleofection, self-renewing medium with ROCKi was added to dilute the P3 buffer. The cell mixture was then taken out and seeded in Matrigel in four wells of a 24-well plate and cultured with self-renewing medium with 10 µM ROCKi for 72 hrs before flow cytometry.

All gRNAs used for gene targeting were pre-tested for efficient DNA cleavage activity using the T7 endonuclease assay and ICE analysis (at least 10% indels, typically 20–30%) before repair template design. Sequences used:

> ACTB 5'-GCTATTCTCGCAGCTCACCA <u>TGG</u>,
> SOX9 5'-CTTGAGGAGGCCTCCCACGA <u>AGG</u>,
> AAVS1 5'-GTCCCCTCCACCCCACAGTG <u>GGG</u>,
> SOX2 N terminal 5'-CGGGCCCGCAGCAAACTTCG <u>GGG</u>,
> SOX2 C terminal 5'-CGGCCCTCACATGTGTGAGA <u>GGG</u>.
> SFTPC 5'-GCGTCCTAGATGTAGTAGAG <u>CGG</u>.
> TP63 5'-TGATGCGCTGTTGCTTATTG <u>CGG</u>.

PAM sequences are underlined.

## Human foetal intestinal culture and generation of *SOX9* reporter organoids

Human foetal gut samples at around 9- to 11-week gestation were obtained with ethical approval (REC-96/085) following maternal consent. Human foetal intestinal organoids were generated from the ileum of the tissue and maintained as previously described (*Ross et al., 2021*). Briefly, the human foetal intestine tissue was placed in a six-well plate and dissociated in a Hank's Balanced Salt Solution (HBSS) medium (14170088, Thermo Fisher Scientific) containing 1.07 Wunsch units/ml of Liberase DH (5401054001, Merck) on a shaking platform at 750 rpm in a 37°C incubator for 15 min. The sample was then mechanically disrupted by pipetting up and down with a P1000 pipette, transferred to an Eppendorf tube and centrifuged for 5 min at 400 g. The supernatant was removed and the pellet was washed three times with Dulbecco's Modified Eagle Medium/F12 (DMEM/F12, 11320033, Thermo Fisher Scientific) by centrifugation. Later, the pellet was re-suspended in Matrigel (Corning, 356231) and seeded in a 48-well plate. The plate was incubated at 37°C for 5 min to solidify the Matrigel and 250 µl of self-renewing medium were added. The self-renewing medium for human foetal intestinal organoids contains Wnt3a conditioned medium

(50%, v/v, Stem Cell Institute, University of Cambridge), R-spondin-1 conditioned medium (20% v/v, Stem Cell Institute, University of Cambridge), Primocin (500 µg/mL, ant-pm-2, InvivoGen), B27 (1:50, 17504044, Thermo Fisher Scientific), nicotinamide (10 mM, N3376, Merck), N-acetylcysteine (1.25 mM, A9165, Merck), A3801 (500 nM, 2939, Tocris), SB202190 (10 µM, S7067, Merck), Murine EGF (50 ng/mL, PMG8041, Thermo Fisher Scientific) and Murine Noggin (100 ng/mL, 250-38, PeproTech). The medium was replaced every 2–3 days and organoids were passaged by mechanical disruption with a P1000 pipette and re-seeded in Matrigel with fresh self-renewing medium every 7–10 days.

For gene targeting, 3 µl spCas9 (4 µg/µl) and 4.32 µl of *SOX9* ssRNA (100 µM, Synthego) were mixed and incubated at RT for a minimum of 10 min in order to form ssRNPs. At the same time, human foetal intestinal organoids were dissociated into single cells, according to the protocol previously described for nucleofection. $6 \times 10^5$ cells were suspended using Lonza Nucleofection P3 buffer, mixed with 6 µg of *SOX9* reporter repair template plasmid. The cell suspension was further mixed with pre-formed Cas9 RNPs and equally distributed into three 20 µl cuvettes. Nucleofection was performed using the program EA125 for human intestinal lung organoids. After nucleofection, intestinal organoid maintenance medium with ROCKi was added to dilute the P3 buffer. The cell mixture was then taken out and seeded in Matrigel in six wells of a 24-well plate and cultured with maintenance medium with 10 µM ROCKi for 1 week before flow cytometry.

EGFP⁺ cells were sorted, harvested and pooled together in 1 × 24-well plate and cultured with intestinal organoid maintenance medium with ROCKi before organoid colonies formed. Individual organoid colonies were picked based on the expression of EGFP observed using EVOS FL system (Thermo Fisher Scientific) microscope and seeded in a 48-well plate to successfully expand them as organoid lines. These organoid lines were then harvested for genomic DNA isolation and genotyping.

## Small-molecule influence on gene targeting efficiency

mEGFP-ACTB gene targeting was performed as previously described. After nucleofection, DMSO (0.6 µl, D2650, Merck), RS-1 (10 µM, R9782, Merck), L755507 (5 µM, SML1362, Merck) or SCR-7 (100 µM, SML1546, Merck) were added to self-renewing medium with ROCKi for 48 hr. Organoid cells were analysed by flow cytometry 72 hr after nucleofection.

## T7 endonuclease assay

To test for site-specific DNA cleavage using the T7 endonuclease assay, organoid cells were harvested 48 hr after nucleofection of ssRNP, tr/cr RNP, plasmid encoding Cas9 and gRNA or WT control organoids. Genomic DNA was extracted using QIAamp Fast DNA Tissue Kit (51404, Qiagen). PCR was performed using PrimeSTAR GXL DNA Polymerase (R050A, Takara) with 20 ng of genomic DNA as template according to the manufacturer's protocol. Forward primer 5'-TTGC-CAATGGGGGATCGCAG-3' and reverse primer 5'-*GCTCGATGGGGGTACTTCAGG*-3' were used for *ACTB* locus amplification. 10 µl of PCR product was then mixed with 1.5 µl 10X NEBuffer 2 (B7002S, NEB) and 1.5 µl of nuclease-free water. The mixture was denatured at 95°C for 10 min, followed by ramp –2°C/s 95°C to 85°C and ramp –0.3°C/s from 85°C to 25°C. 2 µl T7 Endonuclease I (1 U/µl, M0302S, NEB) was added and incubated at 37°C for 1 hr. 2.5% agarose gel was used for electrophoresis.

## ICE analysis for indel production

Genomic DNA was extracted from organoid cells which were harvested 48 hr after nucleofection of ssRNP, tr/cr RNP, plasmid encoding Cas9 and gRNA or WT control organoids using QIAamp Fast DNA Tissue Kit (51404, Qiagen). PCR was performed using PrimeSTAR GXL DNA Polymerase (R050A, Takara) with 20 ng of genomic DNA as template according to the manufacturer's protocol. Forward primer 5'-TTGCCAATGGGGGATCGCAG-3' and reverse primer 5'-GCTCGATGGGGGTACTTCAGG-3' were used for *ACTB* locus amplification. PCR products were cleaned up using Macherey-Nagel NucleoSpin Gel and PCR Clean-up Kit (Macherey-Nagel, 740609.50) and sent for Sanger Sequencing (Department of Biochemistry, University of Cambridge) using reverse primer 5'-GCTCGATGGGG-TACTTCAGG-3'. Sanger sequencing results were compared using ICE online CRISPR editing analysis tool: https://www.synthego.com/products/bioinformatics/crispr-analysis.

## Flow cytometry analysis

Organoid single cells were prepared 72 hr after nucleofection, lentivirus transduction or Lipofectamine transfection. Cells were analysed using Sony SH800S Cell Sorter and FlowJo software (version 10.4).

## Organoid genotyping

Organoids from a single 48-well plate well were used for genomic DNA extraction with QIAamp Fast DNA Tissue Kit (51404, Qiagen) according to the manufacturer's protocol. PCR was performed using PrimeSTAR GXL DNA Polymerase (R050A, Takara) with 20 ng of genomic DNA as template according to the manufacturer's protocol. Primers used are listed in *Supplementary file 2*. For each gene targeting, three randomly picked lines were chosen for further Sanger sequencing. 5′ and 3′ homologous arms of the gene targeting product were amplified using PrimeSTAR GXL DNA Polymerase with the aforementioned primers. PCR products were cleaned up using Macherey-Nagel NucleoSpin Gel and PCR Clean-up Kit (Macherey-Nagel, 740609.50) and sequenced using Sanger sequencing (Department of Biochemistry, University of Cambridge).

## Evaluation of off-target effects

SOX9 gRNA potential off-target sites were identified using the IDT online service: https://eu.idtdna.com/site/order/designtool/index/CRISPR_SEQUENCE. We selected nine potential off-target sites with the least mismatches (three mismatches) from all off-target sites with *NGG* PAM. Primers (*Supplementary file 2*) were designed to amplify the flanking regions of these sites, purified and sent for Sanger sequencing.

## Alveolar differentiation of lung organoids

For alveolar differentiation, organoids were transferred to 300 Pa stiffness poly-ethylene-glycol (PEG) mixture containing RGD adhesion peptides synthesised in Prof. Matthias Lutolf's lab as previously published (*Gjorevski and Lutolf, 2017*). Alveolar differentiation medium, containing N2 (1:100, Thermo Fisher Scientific, 17502-048), B27 (1:50, Thermo Fisher Scientific, 12587-010), N-acetylcysteine (1.25 mM, Merck, A9165), FGF10 (100 ng/ml, PeproTech, 100-26), FGF7 (100 ng/ml, PeproTech, 100-19), CHIR99021 (3 μM, Stem Cell Institute, University of Cambridge), dexamethasone (50 nM, Merck, D4902), cAMP (0.1 mM, Merck, B5386), IBMX (0.1 mM, Merck, I5879) and DAPT (50 μM, Merck, D5942) was used for 6 days.

## Western blot

Organoids were harvested, washed twice with Advanced DMEM/F12 and then twice with PBS before pelleting at 300 g for 5 min. Organoid cell pellets were re-suspended in 100–200 μl of RIPA buffer (Merck, R0278) with protease inhibitor (Thermo Fisher Scientific, 87786) added. Organoid suspension was incubated for 30 min on ice, with strong vortex every 5 min. Cell pieces and debris were removed by centrifugation at 13,000 rpm. Supernatant was harvested. Protein concentration was measured by BCA assay (Thermo Fisher Scientific, 23227). Equal amount of each protein sample was mixed with Sample Buffer (Bio-Rad, 1610747) and beta-mercaptoethanol according to the manufacturer's protocol. Mixture was heated at 95°C for 5 min and cooled down to RT.

Samples were then separated on a 4–12% SDS-PAGE and transferred to nitrocellulose membranes. Membrane was blocked by PBS with 5% BSA for 1 hr at RT. Proteins were detected by incubation with primary antibodies (SOX2, Bio-Techne, AF2018, 1:1000, RRID:AB_355110; α-Tubulin, Merck, T6199, 1:2000, RRID:AB_477583; GFP, Abcam, ab13970, 1:5000, RRID:AB_300798; β-actin, Merck, A3854, 1:50,000, RRID:AB_262011) and secondary antibodies (Donkey anti-Goat IRDye 800CW, Abcam, ab216775, 1:1000, RRID:AB_2893338; Donkey anti-Mouse IRDye 800CW, ab216774, 1:1000, RRID:AB_2893339; IRDye 680LT Donkey anti-Chicken Secondary Antibody, Li-Cor, 925-68028, 1:20,000, RRID:AB_2814923). Protein bands were visualised using Li-Cor Odyssey system.

## Lentivirus infection of organoids for inducible knockdown and activation

Organoids were dissociated into small organoid pieces or single cells using pre-warmed TrypLE Express at 37°C for 5–10 min. Organoid cells were then spun down at 300 g for 5 min. The cell pellet of one 24-well plate was re-suspended in 500 μl self-renewing medium with ROCKi. 20 μl of inducible

CRISPRi or CRISPRa lentivirus was added and mixed well. The mixture was incubated at 37°C over-night. The next morning, organoid cells were pelleted at 300 g for 5 min, washed twice with PBS and seeded in 100 µl Matrigel into two wells of a 24-well plate. Organoid cells were cultured with self-renewing medium with ROCKi for 3 days before dissociating with TrypLE Express into single cells for cell sorting. TagRFP+ cells were sorted using Sony SH800S Cell Sorter and pooled together and seeded in Matrigel at ~ 3000–5000 cells/well of a 24-well plate. Organoid cells were then expanded for around 17 days using self-renewing medium with ROCKi. At this stage, organoids with inducible CRISPRi and CRISPRa system could be frozen as parental lines.

Organoids with inducible CRISPRi and CRISPRa system were broken into small organoid pieces, or single cells, similarly and transduced with gRNA lentivirus (5–10 µl virus/500 µl organoid cell re-suspension). The mixture was incubated overnight, and the next morning organoid cells were pelleted, washed and seeded with Matrigel as described above. After 3 days of culturing with self-renewing medium supplemented with ROCKi, TagRFP and EGFP double-positive cells were sorted and seeded in Matrigel at ~2000–3000 cells/well of a 24-well plate and expanded for 10–17 days before turning on inducible CRISPRa and CRISPRi. Dox (2 µg/ml, Merck, D9891) and TMP (10 nmol/l, Merck, 92131) were supplemented in self-renewing medium accordingly. Medium was refreshed every 48 hr.

For evaluation of inducible CRISPRi using *CD71* as target, single cells were prepared as described above after 5 days of Dox and TMP treatment. Cells were pelleted at 300 g for 5 min and re-suspended in 100 µl PBS with 0.5% BSA and 2 mM EDTA. 2.5 µl of PE/Cy7 anti-human CD71 antibody (BioLegend, 334111, RRID:AB_2563118) was added and incubated at 4°C for 30 min. Cells were then washed with PBS with 0.5% BSA and 2 mM EDTA twice and re-suspended in 300 µl of PBS with 0.5% BSA and 2 mM EDTA for flow cytometry analysis.

## RNA extraction, cDNA synthesis and qRT-PCR

Organoids were harvested and washed twice with Advanced DMEM/F12, before 350 µl of RLT buffer was added to lyse organoids. RNA extraction was performed according to the manufacturer's protocol using RNeasy Mini Kit (Qiagen, 74104) with RNase-Free DNase Set (Qiagen, 79254). RNA concentrations were measured by Nanodrop (Thermo Fisher Scientific). cDNA was synthesised with MultiScribe Reverse Transcriptase (Thermo Fisher Scientific, 4311235) according to the manufacturer's protocol. cDNA was diluted 1:25 and 6 µl was used for one qPCR reaction with PowerUp SYBR Green Master Mix (Thermo Fisher Scientific, A25776). Relative gene expression was calculated using the ΔΔCT method relative to *ACTB* control.

## Plasmid construction

eSpCas9(1.1) was a gift from Feng Zhang (Addgene plasmid # 71814). ACTB gRNA sequence 5′-GCTAT-TCTCGCAGCTCACC-3′ was cloned into the vector using BbsI sites. ACTB repair template AICSDP-15: ACTB-mEGFP was a gift from The Allen Institute for Cell Science (Addgene plasmid # 87425). SOX9 repair template was created by Infusion (638909, Takara) cloning to insert SOX9 5′ and 3′ homologous arms in EasyFusion T2A-H2B-GFP plasmid (a gift from Janet Rossant, Addgene plasmid # 112851). AAVS1 repair template was created by Infusion cloning to swap the CAG promoter and Puromycin resistance cassette in plasmid AICSDP-42: AAVS1-mTagRFPT-CAAX (a gift from The Allen Institute for Cell Science, Addgene plasmid # 107580). SOX2 KO repair template was created by Infusion cloning to insert SOX2 5′ and 3′ homologous arms in EasyFusion T2A-H2B-GFP (a gift from Janet Rossant, Addgene plasmid # 112851). SFTPC targeting repair template was created by Infusion assembly of SFTPC 5′ and 3′ homologous arms together with T2A-Rox-EF1a-Rox-Venus-NLS. TP63 targeting repair template was created by Infusion assembly of TP63 5′ and 3′ homologous arms together with T2A-Rox-EF1a-Rox-Venus-NLS. CMV-Dre-T2A-TagRFP vector was created by Infusion assembly of Dre (a gift from Azim Surani Group, Gurdon Institute, Cambridge) and T2A-TagRFP sequences together. NKX2-1 overexpression vector was created by inserting EF1a-TagRFP-2A-tet3G and tetON-NKX2-1 CDS in two steps cloning using Infusion cloning into a pHAGE backbone. The minimal CMV-GFP(1-10) plasmid was created by Infusion cloning of CMV-GFP(1-10) from pcDNA3.1-GFP(1-10) (a gift from Bo Huang, Addgene plasmid # 70219) into a pUC57 backbone. For testing lentiviral transduction efficiency, CMV-myrAKT-IRES-GFP vector was created by Infusion cloning to insert myrAKT from pCCL-Akt1 (a gift from Bi-Sen Ding, Icahn School of Medicine, Mount Sinai) and IRES sequence into pFP945 (a gift from Frederick Livesey, University College London). The Dox-inducible CRISPRi vector

was created by sub-cloning N-terminal KRAB-dCas9 (a gift from Bruce Conklin, Addgene plasmid # 73498) into the NKX2-1 overexpressing vector using XhoI and BamHI sites. Dox-inducible CRISPRi with DD control vector was created by In-fusion cloning of a C-terminal DHFR sequence into the Dox-inducible CRISPRi vector using BamHI site. The inducible CRISPRa vector was created by sub-cloning dxCas9(3.7)-VPR (a gift from David Liu, Addgene plasmid # 108383) into the NKX2-1 overexpressing vector using XhoI and BamHI sites. The gRNA entry vector was cloned by infusion cloning of a EF1a promoter into pKLV2-U6gRNA5(BbsI)-PGKpuro2ABFP-W vector (a gift from Kosuke Yusa, Addgene plasmid # 67974) using BamHI and EcoRI sites, and then cloned a EGFP-CAAX to swap the EGFP sequence using XhoI and NotI sites. All plasmids have been deposited in Addgene: https://www.addgene.org/browse/article/28216212/.

## Acknowledgements

We thank Dr. Andrew Bassett (Wellcome Sanger Institute) for the Cas9 expressing vector. We thank Prof. Luke Gilbert (UCSF), Prof. Martin Kampmann (UCSF) and Prof. Ruilin Tian (SUSTech) for their advice on CRISPRi and CRISPRa. We would like to acknowledge the Gurdon Institute Imaging Facility for microscopy support.

DS is supported by a Wellcome Trust PhD studentship (109146/Z/15/Z) and the Department of Pathology, University of Cambridge. LDE is supported by the Alzheimer's Research UK Stem Cell Research Centre, funded by the Alborada Trust. VS is supported by a Wellcome Trust PhD studentship (102175/B/13/Z). KL is supported by Basic Science Research Program through the National Research Foundation of Korea (NRF) funded by the Ministry of Education (2018R1A6A3A03012122). MZ was funded by a Medical Research Council (MRC) New Investigator Research Grant (MR/T001917/1). ELR is supported by Medical Research Council (MR/P009581/1). Core support from the Wellcome Trust (203144/Z/16/Z) and Cancer Research UK (C6946/A24843).

## Additional information

### Funding

| Funder | Grant reference number | Author |
|---|---|---|
| Medical Research Council | MR/P009581/1 | Emma L Rawlins |
| Wellcome Trust | PhD Studentship 109146/Z/15/Z | Dawei Sun |
| Alzheimers Research UK Stem Cell Research Centre | | Lewis Evans |
| National Research Foundation of Korea | 2018R1A6A3A03012122 | Kyungtae Lim |
| Wellcome Trust | Core Support for Gurdon Institute 203144/Z/16/Z | Emma L Rawlins |
| Cancer Research UK | Core Support for Gurdon Institute C6946/A24843 | Emma L Rawlins |
| Medical Research Council | New Investigator Research Grant MR/T001917/1 | Matthias Zilbauer |
| Wellcome Trust | PhD studentship 102175/B/13/Z | Vanesa Sokleva |

The funders had no role in study design, data collection and interpretation, or the decision to submit the work for publication.

### Author contributions

Dawei Sun, Conceptualization, Formal analysis, Investigation, Writing - original draft, Writing – review and editing; Lewis Evans, Kyungtae Lim, Investigation, Writing – review and editing; Francesca Perrone, Investigation, Resources, Validation; Vanesa Sokleva, Investigation, Validation; Saba

Rezakhani, Matthias Lutolf, Methodology, Resources; Matthias Zilbauer, Supervision; Emma L Rawlins, Conceptualization, Funding acquisition, Project administration, Supervision, Writing - original draft, Writing – review and editing

## Author ORCIDs
Dawei Sun ⬛ http://orcid.org/0000-0003-1551-4349
Lewis Evans ⬛ http://orcid.org/0000-0001-7279-7651
Kyungtae Lim ⬛ http://orcid.org/0000-0001-6044-2191
Emma L Rawlins ⬛ http://orcid.org/0000-0001-7426-3792

## Decision letter and Author response
Decision letter https://doi.org/10.7554/eLife.67886.sa1
Author response https://doi.org/10.7554/eLife.67886.sa2

## Additional files

### Supplementary files
• Transparent reporting form

• Supplementary file 1. Summary of genetic manipulation efficiency in different organoid lines.

• Supplementary file 2. Primers and gRNA sequences used in this study.

• Supplementary file 3. Raw quantification data. Summary of qRT-PCR results, flow cytometry quantification results of DNA delivery efficiency, small-molecule function and CRISPRi kinetics analysis and ICE analysis results.

• Source data 1. Gels and blots.

### Data availability
All data generated or analysed during this study are included in the manuscript and supporting files.

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
