## [Decision Letter]

**Acceptance summary:**

Sun and colleagues demonstrate the feasibility of using CRISPR-based gene editing techniques applied to tissue-derived human fetal lung organoids. While previous studies have used CRISPR-Cas9 to perform knock-in or knock-out studies in organoids. A major strength of this report is the additional use of CRISPRi and CRISPRa technologies. The work is well done, clearly presented and makes an important contribution to the literature.

**Decision letter after peer review:**

Thank you for submitting your article "A functional genetic toolbox for human tissue-derived organoids" for consideration by *eLife*. Your article has been reviewed by 3 peer reviewers, and the evaluation has been overseen by Marianne Bronner as the Senior and Reviewing Editor. The following individuals involved in review of your submission have agreed to reveal their identity: Tushar J Desai (Reviewer #1); Andrew WIlson (Reviewer #3).

Essential revisions:

Since the manuscript is primarily a methods/tools description, it is essential to include additional description and details to help readers understand the efficiency of the targeting approaches and the expression kinetics of the modified version of dCas9-KRAB. The efficiency and reproducibility of the technology needs to be clarified. Regarding the generalizability of the technique to other kinds of cells, lung or otherwise, while we do not ask you to do new experiments, it is important to clarify the potential application to cells like adult lung stem cells (e.g, HBECs), since access to fetal tissue is very limited in some countries. Please note that the reviewers strongly encourage you to show that your lines can undergo differentiation after genetic manipulation. I refer you to the complete reviews below for details.

*Reviewer #1 (Recommendations for the authors):*

I do have a few criticisms. One is that they only validated the activity in human fetal lung progenitors, so it is not clear whether or not their optimized protocols are 'portable' to other kinds of organoids, even adult lung stem cells. This is somewhat important to know since many biologists don't have ready access to fetal tissues with which to work. I also didn't get a practical sense for how efficient the overall approach is, for instance what is the minimum number of cells needed for success and how much does this vary depending on the allele targeted? There is some information provided in Supplemental Figure 5g which shows high variability. Is the difficulty level predictable in advance with knowledge of the targeted allele or desired manipulation? Did it fail entirely with any organoid line? It's not clear from the way the data is presented, but important to know if a relative non-expert in CRISPR would be likely to have success in using the toolkit.

*Reviewer #2 (Recommendations for the authors):*

1. To overcome the main weakness of the paper, it would be important to show that targeted cells can still differentiate into bronchial airway epithelial (basal and/or goblet cells) after in vitro culture or grafting into the kidney capsule.

2. It would also be important to show whether this differentiation is defective in *SOX2* null cell lines.

3. The authors do a good job in showing that they have obtained correct targeting of the alleles they have selected. However, they do not discuss whether they tested for genome wide, off-target effects. It would be very useful to know whether the efficiency of correct targeting they obtain using homology based repair is higher than in other systems using non homologous targeting. Otherwise, their method is not really as "easy" as it could be.

*Reviewer #3 (Recommendations for the authors):*

To strengthen the manuscript there are points that need to be addressed:

1. As a major focus of the paper is the application of gene editing strategies to human fetal lung organoids, it is important to include the efficiency of each targeting strategy. The efficiency is only stated for the mEGFP-ACTB targeting but it is important for efficiencies to also be shown for the AAVS1-EF1a-mTagRFP-T targeting, *SOX9*-T2A-H2B-EGFP targeting and the SFTPC-T2A-Rox-EF1a-Rox-Venus-NLS targeting to allow readers to understand how robust and reproducible these approaches are.

2. The human fetal lung organoids should express NKX2-1, so it is unclear why overexpression of NKX2-1 is necessary to induce SFTPC expression. The biology underlying this approach should be clearly explained in the manuscript. The SFTPC-T2A-Rox-EF1a-Rox-Venus-NLS reporter imaging in Figure 2G appears to result in cytoplasmic expression of Venus (instead of nuclear which would be expected based on the NLS), and does not appear to be co-expressed in every SFTPC+ cell- please explain this result.

3. To generate a complete *SOX2* knockout, multiple rounds of targeting were required. The efficiency of targeting and re-targeting *SOX2* should be included in the results. In addition, it is noted that *SOX2* is a small gene containing a single exon. What is the feasibility of deleting (and replacing with T2A-H2B-EGFP) a gene larger than *SOX2*? It would seem likely that the design of the homology arms would be increasingly complex and in practice would be less efficient with larger genes. The broader applicability of this approach could be demonstrated by targeting a larger gene in the fetal lung organoids.

4. As this modified version of inducible dCas9-KRAB has not been previously reported, it is important to demonstrate the kinetics of dCas9-KRAB expression following the addition of dox/TMP. It would also be useful to compare the kinetics of dCas9-KRAB expression between the original TetON-dCas9-KRAB system with this modified TetON-dCas9-KRAB-DHFR system, potentially by qPCR to quantify expression of dCas9 over time. This approach could also be applied to compare untargeted lines to lines targeted with the original TetON-dCas9-KRAB to more clearly establish that basal leak does, in fact, occur. Lastly, it is necessary to show how tunable this CRISPRi system is, for instance when the dox/TMP is removed how quickly is gene expression restored?

5. The CRISPRi-induced knock-down of *SOX2* results in loss of ~75% of mRNA but appears to result in 100% loss of protein by Western blot. Is this consistent across replicates?

6. In some experiments, the cells are cultured with ROCK inhibitor for up to 17 days. Was it tested whether ROCK inhibitor could be removed after the single cell stage, given that long-term culture in ROCK inhibitor can affect metabolism (PMID 28165055)?

---

## [Author Response]

Essential revisions:Since the manuscript is primarily a methods/tools description, it is essential to include additional description and details to help readers understand the efficiency of the targeting approaches and the expression kinetics of the modified version of dCas9-KRAB. The efficiency and reproducibility of the technology needs to be clarified. Regarding the generalizability of the technique to other kinds of cells, lung or otherwise, while we do not ask you to do new experiments, it is important to clarify the potential application to cells like adult lung stem cells (e.g, HBECs), since access to fetal tissue is very limited in some countries. Please note that the reviewers strongly encourage you to show that your lines can undergo differentiation after genetic manipulation. I refer you to the complete reviews below for details.

We are grateful for the absence of a time limit for providing a revised version of the manuscript. We felt that we needed to perform additional experiments to fully address some of the reviewer’s comments and access to tissue culture has been extremely limited this year due to the requirements to maintain social distancing.

We thank the editor’s effort to put this manuscript into the revision process and to summarise the reviewers’ comments in the essential revisions section. We hope that the new version of manuscript both addresses the Essential Revisions and also demonstrates our efforts to address the comments/concerns from all reviewers.

(1) In order to help the readers understand better the efficiency of the Easytag Workflow, we have generated a more thorough table (New supplementary table: Supplementary File 1), summarising the organoid lines tested for different applications, numbers of cells used and the number of lines which passed genotyping. We believe that these details will help the readers to have a practical sense about using the Organoid Easytag workflow.

In addition, we summarized the line-to-line variability of our CRISPRi and CRISPRa approaches in the same table.

(2) Instead of only quantifying dCas9-KRAB-DHFR to show the kinetics of the inducible CRISPRi system as suggested by the reviewers, we have systematically sampled and analysed the kinetics of the knockdown effect for the inducible CRISPRi system using CD71 protein and *SOX2* mRNA as read-outs (New figure: Figure 3—figure supplement 2). We reasoned that this will be more informative for the reader than simply knowing the expression level of *dCas9-KRAB-DHFR* itself, which is likely to be only determined by the Dox inducible system. This analysis has revealed the quick response of the system in knocking down *SOX2* mRNA expression on Day 1 and a gradual knockdown effect on CD71 protein from Day 1 to Day5 after Dox and TMP treatment (Figure 3—figure supplement 2 ).

Furthermore, we have explored the reversibility of the inducible CRISPRi system. We observed that the CRISPRi knockdown effect recovered within one-two weeks after Dox and TMP removal (Figure 3—figure supplement 2 ). We further validated that after the CRISPRi knockdown was recovered, the system remained fully functional and knock-down could be induced again by further Dox and TMP treatment (Figure 3—figure supplement 2c).

This efficient knockdown effect with full reversibility demonstrates that our adaptation of inducible CRISPRi is a powerful system to study gene function in human tissue derived organoid systems. It will be applicable to any organoid line in which methods for lentiviral transduction have been optimised.

(3) In order to further generalise the Organoid Easytag workflow, we have successfully generated *SOX9* reporter organoid lines in the human foetal intestinal organoid system (New figures: Figure 2e-f). This demonstrates that the technique is potentially generalisable to other organoid systems.

Additionally, we have tested using the inducible CRISPRi system on human bronchial epithelial cells (HBECs) as suggested by the reviewers. However, the HBECs were very sensitive to lentiviral transduction. We observed massive cell death after transduction and cell passaging, even when cells were only transduced with the gRNA lentivirus (vector diagram in Figure 3a, lower panel). We believe that setting up the inducible CRISPRi system in HBECs would require further optimisation for DNA delivery, probably via Nucleofection (Siddiqui et al., 2021) plus a PiggyBac transposon system for integration.

However, it is clear that the inducible CRISPRi workflow will be easy to adopt for use in any cell, or organoid, line in which lentiviral transductions have been optimised. The potential for generalisation is now clearly stated at the end of the manuscript:

Page 9: We have established a complete organoid genetic toolbox for gene targeting, reporter generation, controllable gene knockouts, inducible gene knockdown and gene activation in the human foetal lung organoid system (efficiency details provided in Supplementary File 1). We envision that these tools can be easily adapted for organoid systems derived from other tissues that are suitable to be nucleofected, or transduced by lentivirus.

(4) In order to show that gene-targeted organoid lines can still undergo differentiation, we have differentiated our reporter organoid lines to the alveolar lineage (New figure: Figure 2—figure supplement 1g).

Reviewer #1 (Recommendations for the authors):I do have a few criticisms. One is that they only validated the activity in human fetal lung progenitors, so it is not clear whether or not their optimized protocols are 'portable' to other kinds of organoids, even adult lung stem cells. This is somewhat important to know since many biologists don't have ready access to fetal tissues with which to work. I also didn't get a practical sense for how efficient the overall approach is, for instance what is the minimum number of cells needed for success and how much does this vary depending on the allele targeted? There is some information provided in Supplemental Figure 5g which shows high variability. Is the difficulty level predictable in advance with knowledge of the targeted allele or desired manipulation? Did it fail entirely with any organoid line? It's not clear from the way the data is presented, but important to know if a relative non-expert in CRISPR would be likely to have success in using the toolkit.

We thank the reviewer for positive comments on our manuscript. A major concern here regards making the data more accessible to the non-expert in CRISPR to have a good sense of using the toolkit and the variability of the method regarding different loci and organoid lines.

We have now generated a new table Supplementary File 1 to summarise the cell lines tested for the Easytag workflow, inducible CRISPRi and inducible CRISPRa. Plus the cell number used for each targeting experiment, numbers of new organoid lines, and correctly targeted lines, derived for each organoid line tested.

From the table, the questions raised by Reviewer#1 can be answered clearly: minimum cell number used was 400K organoid cells and it varied among the different targets; there were organoid lines that were not very suitable for gene targeting, e.g. BRC1929 and BRC1938, which reflects the variable nature of these primary human cells. We believe that this will help the non-experts to have a practical sense of using the toolkit.

Regarding Reviewer #1’s question about predicting targeting success in advance. It’s generally quite difficult to predict the success in advance for gene targeting, even in established systems like mouse ESCs. We have developed a simple empirical test for this. We reasoned that an efficient DNA cleavage is the pre-requisite for HDR to be triggered. Therefore, we pre-tested the gRNA cleavage ability (at least 2 gRNAs/locus) as a predictor for all gene targeting that we have performed. This cleavage assessment was performed using the T7 endonuclease assay and ICE analysis (Figure 1—figure supplement 1c-d). This test also facilitates the next step repair of template design, where a mutation in the PAM sequence of the repair template is required to avoid Cas9 cleavage of the repair templates. This method has given us a good prediction that the targeting would work efficiently in most cases, except targeting of the *SFTPC* locus which was inefficient (Supplementary File 1). Therefore, we believe the activity of gRNA on DNA cleavage can be a good empirical prediction for the efficiency of gene targeting.

We have now specified the pre-testing of gRNA cleavage activity prior to beginning a targeting experiment in the Methods section.

Page 19: All gRNAs used for gene targeting were pre-tested for efficient DNA cleavage activity using the T7 endonuclease assay and ICE analysis (at least 10% indels, typically 20-30%) before repair template design.

Reviewer #2 (Recommendations for the authors):1. To overcome the main weakness of the paper, it would be important to show that targeted cells can still differentiate into bronchial airway epithelial (basal and/or goblet cells) after in vitro culture or grafting into the kidney capsule.

We agree with the Reviewer that it would be important to show that the targeted organoids can still be differentiated. Therefore, we have now differentiated our *SOX9* reporter line to the alveolar type 2 lineage (New Figure: Figure 2—figure supplement 1g). We have also tested the dual SMAD inhibition approach recently reported for basal cell differentiation (Miller et al., 2020). However, the massive cell death, which is consistent with the same report, led us to think this would require further optimisation for applying this approach on our organoids which are derived from tissue at an earlier gestation stage. Given that we achieved alveolar differentiation in targeted organoids, we hope the Reviewer would agree with us that further optimisation of differentiation protocols is beyond the scope of this manuscript.

2. It would also be important to show whether this differentiation is defective in SOX2 null cell lines.

We agree with the Reviewer this would be interesting to test and we are actually also very interested in this question given that *SOX2* has been implicated in mouse research to be important for bronchiolar lineage formation (Ochieng et al., 2014; Que et al., 2009). However, as mentioned above, further optimisation of the basal cell differentiation protocol is required for our organoid cells, so we feel this is beyond the scope of this technically-focused manuscript.

3. The authors do a good job in showing that they have obtained correct targeting of the alleles they have selected. However, they do not discuss whether they tested for genome wide, off-target effects. It would be very useful to know whether the efficiency of correct targeting they obtain using homology based repair is higher than in other systems using non homologous targeting. Otherwise, their method is not really as "easy" as it could be.

We agree that testing genome-wide, off-target effects would strengthen our manuscript. As a proof of concept, we analysed potential off-target sites for the *SOX9* gRNA used in our study and selected a list of potential off-target sites (New figures: Figure 2—figure supplement 1e-f). We focused on the most probable off-target sites which all had 3 mismatches to the *SOX9* gRNA. We amplified and sequenced neighbouring regions of the only off-target site targeting an exon (*ST3GAL1* gene) together with the other 8 off-target sites. We observed no indels within at least ~200 bp flanking the gRNA targeting sites at all sites tested in 3 *SOX9*-targeted organoid lines (New figures: Figure 2—figure supplement 1e-f), suggesting minimal off-target effects.

We have compared the efficiency of correctly targeted lines derived using the Easytag workflow with the CRISPR-HOT approach. The percentage of in-frame knock-ins is 40% and 33% for *TUBB* and *CDH1* genes respectively in CRISPR-HOT (Artegiani et al., 2020). This is comparable with most of the Organoid Easytag workflow targeting efficiencies ranging from 39%-66%, except for the low efficiency for the *SFTPC* locus (6.25%). However, we would like to emphasize we cannot easily conclude which one is ‘more efficient’ as the targeting efficiency is dependent on many factors, including the cell type used, target sites, gRNA used and the length of the inserted sequences.

Reviewer #3 (Recommendations for the authors):To strengthen the manuscript there are points that need to be addressed:1. As a major focus of the paper is the application of gene editing strategies to human fetal lung organoids, it is important to include the efficiency of each targeting strategy. The efficiency is only stated for the mEGFP-ACTB targeting but it is important for efficiencies to also be shown for the AAVS1-EF1a-mTagRFP-T targeting, SOX9-T2A-H2B-EGFP targeting and the SFTPC-T2A-Rox-EF1a-Rox-Venus-NLS targeting to allow readers to understand how robust and reproducible these approaches are.

This comment is in line with the comments from Reviewer #1. We agree with the Reviewer that it would be necessary to give the readers a practical sense on how reproducible our approaches are. Therefore, we have summarised the details of each targeting experiment, including cell lines tested, cell number used and correct colonies genotyped in the Supplementary File (New table: Supplementary File 1).

2. The human fetal lung organoids should express NKX2-1, so it is unclear why overexpression of NKX2-1 is necessary to induce SFTPC expression. The biology underlying this approach should be clearly explained in the manuscript. The SFTPC-T2A-Rox-EF1a-Rox-Venus-NLS reporter imaging in Figure 2G appears to result in cytoplasmic expression of Venus (instead of nuclear which would be expected based on the NLS), and does not appear to be co-expressed in every SFTPC+ cell- please explain this result.

NKX2-1 direct binding upstream of *SFTPC* and promotion of *SFTPC* expression has recently been reported in a new preprint from our lab (Lim et al., 2021). Lim et al. show that NKX2-1 is initially expressed at relatively low levels in the human embryonic/early foetal lung (the stage at which the organoids in this manuscript were derived) and is upregulated as alveolar specification occurs where it directly activates *SFTPC* transcription. Here, we used NKX2-1 overexpression to activate *SFTPC* expression. (We now reference the preprint in the text of this manuscript.)

In regard to the cytoplasmic expression of Venus, we were also surprised to observe this result. Venus was nuclear prior to Dre recombinase expression (as shown in Figure 2—figure supplement 2c). We speculated that the T2A sequence was not efficiently cleaved to separate SFTPC and Venus-NLS resulting in Venus’ cytoplasmic location. We have now tested protein size in a western blot and visualised a larger band (40-50 kDa) following Dre transduction, compared with the original Venus-NLS expressed after the first step of gene targeting (New figure: Figure 2—figure supplement 2f). Given that pro-SFTPC is 21kDa and Venus is 27kDa, the size of the band correlated well with the predicted size of the fusion protein, suggesting that inefficient T2A cleavage might be the cause of the cytoplasmic protein localization. We believe that this is an unlucky chance and the final molecule happens to have an unfavourable configuration for T2A cleavage. Nevertheless, the system works in principle and Venus is visible in cells with a high level of pro-SFTPC protein.

3. To generate a complete SOX2 knockout, multiple rounds of targeting were required. The efficiency of targeting and re-targeting SOX2 should be included in the results. In addition, it is noted that SOX2 is a small gene containing a single exon. What is the feasibility of deleting (and replacing with T2A-H2B-EGFP) a gene larger than SOX2? It would seem likely that the design of the homology arms would be increasingly complex and in practice would be less efficient with larger genes. The broader applicability of this approach could be demonstrated by targeting a larger gene in the fetal lung organoids.

We have now included the efficiency of *SOX2* targeting and retargeting in the Supplementary File (New table: Supplementary File 1).

We agree with the Reviewer’s concern that there will be a size limit to the dual gRNA approach and that this still remains to be tested. We also agree with the Reviewer’s comment that *SOX2* is a gene with single exon, which is likely relatively easy for CDS swapping.

However, we believe that even for a more complex locus, the repair template design would still be as simple as just using the 800-1000 nt genomic sequence upstream and downstream of the 1^st^ and 2^nd^ gRNAs, respectively. Alternatively, we could use the two-gRNA approaches as demonstrated in this manuscript to swap a functional domain of a specific gene to generate a functional knockout, e.g. a *SOX9* functional KO could be generated by swapping the transactivation domain located within the 3^rd^ exon with EGFP. This would be analogous to “floxing” a single exon of a gene in mouse ESCs to generate conditional KO mice. In this example, the repair template design would be very similar to the design of the *SOX2* KO shown in this manuscript. Finally, we would like to mention that there are other strategies with using a single gRNA to insert an EGFP-polyA sequence in frame in an early exon that would also be sufficient for KO purposes, and the repair template design is very similar to generating a fusion protein.

4. As this modified version of inducible dCas9-KRAB has not been previously reported, it is important to demonstrate the kinetics of dCas9-KRAB expression following the addition of dox/TMP. It would also be useful to compare the kinetics of dCas9-KRAB expression between the original TetON-dCas9-KRAB system with this modified TetON-dCas9-KRAB-DHFR system, potentially by qPCR to quantify expression of dCas9 over time. This approach could also be applied to compare untargeted lines to lines targeted with the original TetON-dCas9-KRAB to more clearly establish that basal leak does, in fact, occur. Lastly, it is necessary to show how tunable this CRISPRi system is, for instance when the dox/TMP is removed how quickly is gene expression restored?

We agree with the Reviewer that this modified version of dCas9-KRAB has not been previously reported. However, the concept of using destabilising domains to control the leaky effect of the tetON system has been established in the literature (Balboa et al., 2015).

In order to validate the basal leakiness, we have used RT-qPCR to quantify the *KRAB-dCas9* expression level without DHFR control and observed its basal expression at a level similar to the *MYCN* expression level (New figure: Figure 3—figure supplement 1e), which is a tip progenitor cell enriched transcription factor (Nikolić et al., 2017). This confirms background expression of *KRAB-dCas9*.

Since the DHFR control is at protein level, we reasoned that RT-qPCR testing *KRAB-dCas9-DHFR* is not sufficient to reveal kinetics, as only its RNA level is controlled by Dox. Therefore, in order to show clearly the kinetics of the inducible system, we have used CD71 and *SOX2* to probe the knockdown kinetics functionally (New figure: Figure 3—figure supplement 2). The knockdown effect can be observed both at RNA and protein levels on Day 1. This is in line with the concept that the background KRAB-dCas9-DHFR (resulting from leaky transcription) can be activated quickly after TMP is administered.

In regard to the reversibility of the inducible CRISPRi system, we have tested this systematically using CD71 and *SOX2* at both protein and RNA level respectively (New figure: Figure 3—figure supplement 2). In both cases, we observed the recovery of gene expression (New figures: Figure 3—figure supplement 2b; 2d) within 1-2 weeks, which is in line with a recent publication (Nuñez et al., 2021). We further confirmed that the inducible CRISPRi system was still fully functional after recovery by observing further silencing upon repeated Dox and TMP treatment (New figure: Figure 3—figure supplement 2c).

5. The CRISPRi-induced knock-down of SOX2 results in loss of ~75% of mRNA but appears to result in 100% loss of protein by Western blot. Is this consistent across replicates?

We have now included a second replicate using a different *SOX2* gRNA and observed a similar KD effect compared with the first replicate in the WB. We think the discrepancy between RT-qPCR and WB could be due to RT-qPCR being more sensitive.

6. In some experiments, the cells are cultured with ROCK inhibitor for up to 17 days. Was it tested whether ROCK inhibitor could be removed after the single cell stage, given that long-term culture in ROCK inhibitor can affect metabolism (PMID 28165055)?

We agree with the Reviewer’s concern about using ROCK inhibitor. The ROCK inhibitor could be removed after the single cell stage. However, organoid cells grow better with ROCK inhibitor and inclusion of the ROCK inhibitor made picking single colonies easier. We would like to stress that in all these experiments we have used a non-targeting control which was treated identically to the experimental group, so we believe that ROCK inhibitor treatment will not greatly affect the conclusions that we made.